# Polyoxometalate Dicationic Ionic Liquids as Catalyst for Extractive Coupled Catalytic Oxidative Desulfurization

**Jingwen Li, Yanwen Guo, Junjun Tan and Bing Hu \***

School of Materials and Chemical Engineering, Hubei University of Technology, Wuhan 430068, China; lijingwen@hbut.edu.cn (J.L.); guoyanwen@hbut.edu.cn (Y.G.); tanjunjun2011@hbut.edu.cn (J.T.)

\* Correspondence: hubing@hbut.edu.cn; Tel.: +86 13667257353

**Abstract:** Wettability is an important factor affecting the performance of catalytic oxidative desulfurization. In order to develop an efficient catalyst for the extractive coupled catalytic oxidative desulfurization (ECODS) of fuel oil by $H_2O_2$ and acetonitrile, a novel family of imidazole-based polyoxometalate dicationic ionic liquids (POM-DILs) $[C_n(MIM)_2]PW_{12}O_{40}$ (n = 2,4,6) was synthesized by modifying phosphotungstic acid ($H_3PW_{12}O_{40}$) with double imidazole ionic liquid. These kinds of catalysts have good dispersity in oil phase and $H_2O_2$, which is conducive to the deep desulfurization of fuel oil. The catalytic performance of the catalysts was studied under different conditions by removing aromatic sulfur compound dibenzothiophene (DBT) from model oil. Results showed that $[C_2(MIM)_2]PW_{12}O_{40}$ had excellent desulfurization efficiency, and more than 98% of DBT was removed under optimum conditions. In addition, it also exhibited good recyclability, and activity with no significant decline after seven reaction cycles. Meanwhile, dibenzothiophene sulfone ($DBTO_2$), the only oxidation product of DBT, was confirmed by Gas Chromatography-Mass Spectrometry (GC-MS), and a possible mechanism of the ECODS process was proposed.

**Keywords:** polyoxometalate; dicationic ionic liquids; extraction; oxidative desulfurization; dibenzothiophene

## 1. Introduction

With the development of the economy, traditional fossil fuels still occupy a large proportion of supply and demand in the market [1]. Some sulfur compounds contained in fuel oil, such as mercaptan, thioether, thiophene and their derivatives, will produce sulfur oxides during combustion, which can lead to a series of environmental problems such as acid rain and haze [2–4]. Therefore, many countries are constantly strengthening the control standards of sulfur content in fuel oil. Improving the technology to produce high quality fuel oil in accordance with the standards has become a top priority for refineries [5,6]. Hydrodesulfurization (HDS) is the most mature technique and has been applied in industry [7–10]. It can efficiently eliminate aliphatic sulfur compounds, such as mercaptan and thioether. However, in addition to the harsh operation conditions, HDS is not effective for removing aromatic sulfur compounds and their derivatives with steric hindrance [11,12]. In this context, as a nonhydrodesulfurization technology that can achieve deep desulfurization of fuel oil under mild conditions, the ECODS process has become a main focus due to its simplicity and effectiveness [13–16]. Although various types of solvents and oxidants have been used in the desulfurization process, and also play an important role, the biggest challenge for a successful ECODS process is to use catalysts with high activity.

Polyoxometalates (POMs), represented by $H_3PW_{12}O_{40}$, have been widely used as the catalysts for oxidative desulfurization under mild conditions of the model oil system due to their strong Bronsteic acidities and redox properties [17–20]. On the other hand,

ionic liquids (ILs), as a prominent catalyst/extractant with low vapor pressure, good thermal stability, recyclability and environmental friendliness, are also usually used in desulfurization reactions [21]. However, the shortcomings of POMs and ILs are the main obstacles to their industrial application. For example, the small specific surface area of POMs ($<10 m^2/g$) makes their catalytic activity low [22], and the liquid properties of ILs make them difficult to separate and recover. In order to solve those problem, according to the characteristics that specific ILs with different properties can be designed by the combination of different cations and anions [23], and the catalytic performance of POMs can be regulated by the electrostatic interaction and hydrogen bonding between the cations of specific ionic liquids and the anions of POMs [24], a new type of POM-IL with different physical and chemical properties, which is formed by the combination of heteropolyanions and organic cations, has attracted widespread attention. Huang et al. synthesized a heteropolyanionic-based ionic liquid catalyst [(3-sulfonic acid) propylpyridine]$_3$PW$_{12}$O$_{40}$·2H$_2$O ([PSPy]$_3$PW$_{12}$O$_{40}$·2H$_2$O) by the reaction of N-Propanesulfone pyridinium with an aqueous solution of H$_3$PW$_{12}$O$_{40}$, which showed high catalytic activity and excellent recyclability in the oxidative desulfurization of fuel oil [25]. Our groups successively synthesized a kind of POM-IL, [Hmim]$_5$PMo$_{10}$V$_2$O$_{40}$ [26], [C$_3$H$_3$N$_2$(CH$_3$)(C$_n$H$_{2n}$)]$_5$PMo$_{10}$V$_2$O$_{40}$ ([C$_n$mim]PMoV n = 2, 4 and 6) [27], by the reaction of molybdovanadophosphoric acid (H$_5$PMo$_{10}$O$_{40}$) with N-methylimidazole and imidazole bromides, respectively, and applied it as a catalyst to the desulfurization process with H$_2$O$_2$ as the oxidant. The results showed that 99.1% and 100% of dibenzothiophene (DBT) in the model oil are removed, respectively, and the catalytic activity of POM-ILs decreased slightly after six cycles. However, these systems still need more catalysts and a relatively long reaction time to achieve ideal desulfurization efficiency. Therefore, it is necessary to find other methods to improve the economic applicability and effectiveness of the catalysts. In recent years, many other types of POM-ILs have been used as catalysts for oxidative desulfurization, such as [C$_{11}$H$_9$N(CH$_2$)$_4$SO$_3$H]$_3$PW$_{12}$O$_{40}$ (PhPyBs-PW) [28], [3-(pyridine-1-ium-1-yl)propane-1-sulfonate]$_3$(NH$_4$)$_3$Mo$_7$O$_{24}$·4H$_2$O ([PyPS]$_3$(NH$_4$)$_3$Mo$_7$O$_{24}$) [C$_6$H$_5$NO$_2$CH$_2$(CH$_2$)$_2$CH$_3$]$_7$PMo$_{12}$O$_{40}$ ([29] and (NKBu)$_7$PMo$_{12}$O$_{42}$) [30], which can effectively improve the desulfurization efficiency in a short period of time with a small amount of catalyst. However, it is rarely reported that POM-based dicationic ionic liquids with higher thermal stability, good wettability and high activity are used as catalysts for oxidative desulfurization. DILs are a new type of ionic liquid compound with higher stability and lower toxicity, which consists of two monomers linked by alkyl or aryl groups [31–34]. Compared with the traditional monocationic ILs, DILs have a larger cationic volume, which makes the π-π interaction between cations and aromatic sulfides stronger, and can effectively remove aromatic sulfides in fuel oil. In addition, through electrostatic interaction and hydrogen bonding, the double cation can be well connected with the anion, so as to improve the overall catalytic activity of the catalyst. Due to their unique properties, DILs have been successfully used and achieved ideal effects in esterification, supercapacitor, biodiesel catalysis and extractive desulfurization as eutectic solvents/catalysts, electrolyte additives, catalysts and extractant, respectively [35–38]. In our group's recent research results, a series of novel binuclear magnetic ionic liquids (MILs) [C$_n$(MIM)$_2$]Cl$_2$/mFeCl$_3$ (n = 2, 4 or 6 and m = 1, 2 or 3) were synthesized and used as catalysts for the desulfurization with oxidant H$_2$O$_2$ and extractant acetonitrile [39]. The results showed that 97.07% desulfurization efficiency can be achieved in 10 min, showing ultrahigh catalytic activity of MILs.

Inspired by the above research, we are deeply interested in the preparation of novel POM-DILs and their application in the field of catalytic oxidative desulfurization. In this work, in order to give full play to the advantages of POMs and DILs, we prepared a new kind of POM-DIL catalyst [C$_n$(MIM)$_2$]PW$_{12}$O$_{40}$ (n = 2, 4, 6) with H$_3$PW$_{12}$O$_{40}$ modified by imidazole-based DILs, and the catalyst was further applied in ECODS system, which was constructed with H$_2$O$_2$ as the oxidant and acetonitrile as the extractant. The optimum reaction conditions and parameters were determined. In addition to their high

thermal stability, the catalysts also showed high catalytic activity for the removal of DBT from model oil due to their excellent wettability. At the same time, the recycling performance of the catalyst was also explored. Finally, based on the corresponding characterization results, the possible mechanism of the desulfurization process was proposed.

## 2. Results and Discussion

### 2.1. Characterization of Catalyst

Fourier transform infrared spectroscopy (FT-IR ) is a suitable technique to prove the success of catalyst synthesis. The FT-IR spectra (wavelength range from 4000 to 500 cm$^{-1}$) of catalysts [C$_2$(MIM)$_2$]PW$_{12}$O$_{40}$, [C$_4$(MIM)$_2$]PW$_{12}$O$_{40}$, [C$_6$(MIM)$_2$]PW$_{12}$O$_{40}$ are shown in Figure 1a. The absorption peak near 2950 cm$^{-1}$ was attributed to the C-H stretching vibration of imidazole ring. The stretching vibration absorption peaks around 1630 and 1564 cm$^{-1}$ were attributed to C=C and C=N skeleton vibrations on the aromatic ring, respectively. The peaks at 1463, 1408 cm$^{-1}$, 1341 and 1253 cm$^{-1}$ were related to C-N heterocycles. In the range of 870-1100 cm$^{-1}$, four characteristic absorption peaks of Keggin structure were observed in all three catalysts, which were 1084 (P-O), 983 (W=O), 898 (W-O$_c$-W corner-sharing) and 800 cm$^{-1}$ (W-O$_e$-W edge-sharing), respectively. This was consistent with the characteristic peaks of H$_3$PW$_{12}$O$_{40}$ in Figure 1b, which showed that the catalysts still had the Keggin structure of PW$_{12}$O$_{40}{}^{3-}$.

X-ray diffraction (XRD) also provided strong evidence to support the successful synthesis of catalysts. Compared with the XRD pattern (2θ from 5° to 50°) of catalysts (Figure 1c) and H$_3$PW$_{12}$O$_{40}$ (Figure 1d), some diffractive peaks of the three catalysts were obtained at near 10°, 21° and 32.8°. This was due to the disappearance of coordination water H$_5$O$_2{}^+$ and H$_3$O$^+$ which interact with the anions of the Keggin structure by replacing the secondary structure proton of H$_3$PW$_{12}$O$_{40}$ with [C$_n$(MIM)$_2$]$^{2+}$ (n = 2, 4, 6).

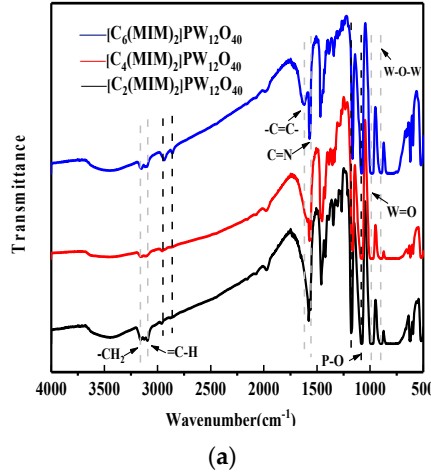

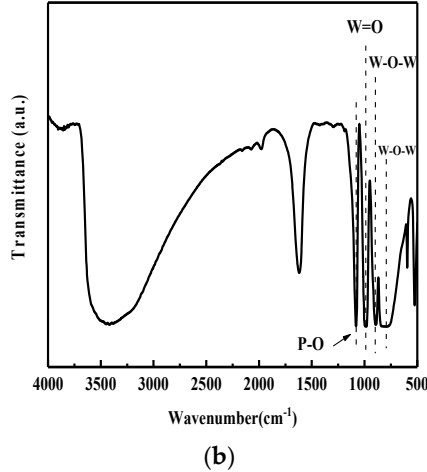

(a)            (b)

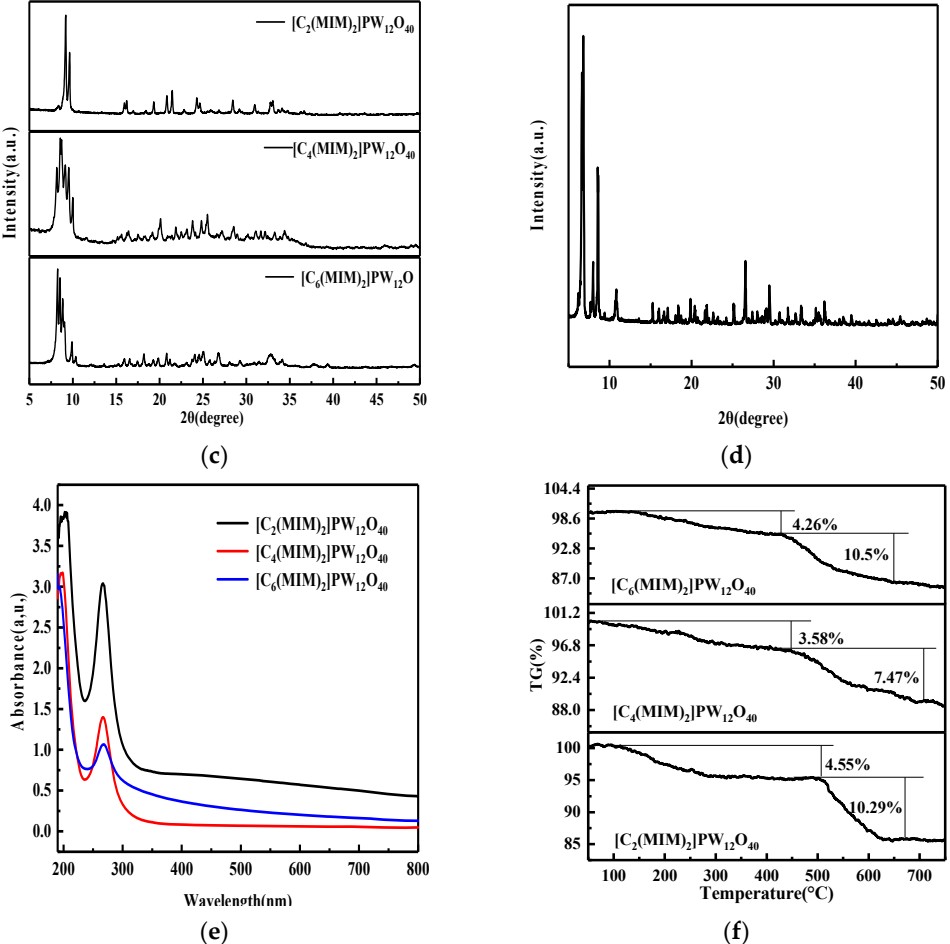

**Figure 1.** Characterizations of the samples. (**a**) FT-IR spectra of catalysts; (**b**) FT-IR spectra of $H_3PW_{12}O_{40}$; (**c**) XRD patterns of catalysts; (**d**) XRD patterns of $H_3PW_{12}O_{40}$; (**e**) Ultraviolet visible (UV-vis) spectra of catalysts; (**f**) Thermogravimetric (TG) curve of catalysts.

UV-vis spectroscopy is a rapid and accurate method to determine the molecular structure of organic compounds and the charge transfer behavior of catalysts. The UV-vis spectra of the catalysts are shown in Figure 1e. There were two characteristic peaks in the range of 190-400 nm near the ultraviolet region, which were related to the electronic properties of the center metal atoms in the anions of the catalyst structure. This structure was similar to $[PW_{12}O_{40}]^{3-}$ [40]. The absorption peaks at 203, 196 and 191 nm of the three catalysts were caused by O→P transition, and the strong absorption peaks at 266, 266 and 267 nm of the three catalysts were considered to charge the transfer of metal atoms ($O^{2-}$→$W^{6+}$), where W atoms were located in W-$O_e$-W intrabridges between edge-sharing $WO_6$ octahedra in the Keggin units.

By recording the TG curve of the synthesized catalysts, the thermal stability of the catalysts can be clearly displayed in Figure 1f. The first mass loss occurred at 100 °C, which was caused by the disappearance of physical water and crystallization water in the catalyst. With the increase in temperature, the weight loss within the range of 300 to 800 °C was related to the decomposition of catalysts. The $[C_n(MIM)_2]^{2+}$ cation was decomposed first; the initial decomposition temperature of the three POM-DIL catalysts was 495 °C for $[C_2(MIM)_2]PW_{12}O_{40}$, 420 °C for $[C_4(MIM)_2]PW_{12}O_{40}$ and 427 °C for $[C_6(MIM)_2]PW_{12}O_{40}$. Then, the $PW_{12}O_{40}{}^{3-}$ anion was decomposed at about 600 °C [25]. From the results, it can be concluded that the reasons for the high thermal stability of POM-DIL catalysts were not only the cation symmetric structure of the double imidazole

ring but also the Keggin structure of the anion. At the same time, the carbon chain length of cations had no obvious effect on the thermal stability of the catalysts.

In order to further accurately determine the moisture in the samples, the content of free water in the samples was determined by the Karl Fischer Titrator (KFT) Ipol method with the Karl Fischer Moisture Titrator (870 KF Titrino plus), and the results are listed in Table 1. From the test results, the moisture content in the sample was extremely small, and the purity of each sample could reach 99%.

**Table 1.** The moisture in the samples.

| Samples | First determination (%) | Second determination (%) | Average value (%) |
|---|---|---|---|
| $[C_2(MIM)_2]PW_{12}O_{40}$ | 0.43 | 0.47 | 0.45 |
| $[C_4(MIM)_2]PW_{12}O_{40}$ | 0.83 | 1.32 | 1.08 |
| $[C_6(MIM)_2]PW_{12}O_{40}$ | 0.19 | 0.19 | 0.19 |

X-ray photoelectron spectroscopy (XPS) characterization is a powerful technique to determine the composition, content and molecular structure of catalysts [41]. The composition of the catalyst $[C_2(MIM)_2]PW_{12}O_{40}$ was analyzed by XPS, as can be seen from the survey spectrum of the sample (Figure 2a), the catalyst was mainly composed of C, N, O, P and W elements. The contents of each element are listed in Table 2. The results showed that $[C_2(MIM)_2]PW_{12}O_{40}$ was mainly composed of C and O elements. At the same time, the higher O content indicated that there were abundant oxygen-containing groups in the catalyst, which was consistent with the characterization results of FT-IR. The XPS spectrum of C1s (Figure 2b) can be fitted into three peaks with binding energies of 285.3, 284.5 and 283.3 eV, respectively, which were attributed to C=N-C, C-C/C=C and C=N. In the XPS spectra of N 1s, O 1s and P 2s (Figure 2c-e), the corresponding binding energies were 400, 529.5 and 132.9 eV, which correspond to C-N, $O^{2-}$ and P-O, respectively. The two peaks at 36.7 and 34.6 eV (Figure 3f) were attributed to W $4f_{5/2}$ and W $4f_{7/2}$. Compared with the existing literature [42,43], due to the electrostatic interaction between the cation and the anion, significant electron shift was observed in the W4f spectra. The results showed that the surface electron density of the catalysts was effectively enhanced, which was conducive to the formation of hydroxyl radicals, thus improving the overall ECODS performance.

**Table 2.** Element composition and content of $[C_2(MIM)_2]PW_{12}O_{40}$.

| Elements | Atomic (%) |
|---|---|
| C | 27.29 |
| N | 10.03 |
| O | 47.93 |
| P | 3.12 |
| W | 11.63 |

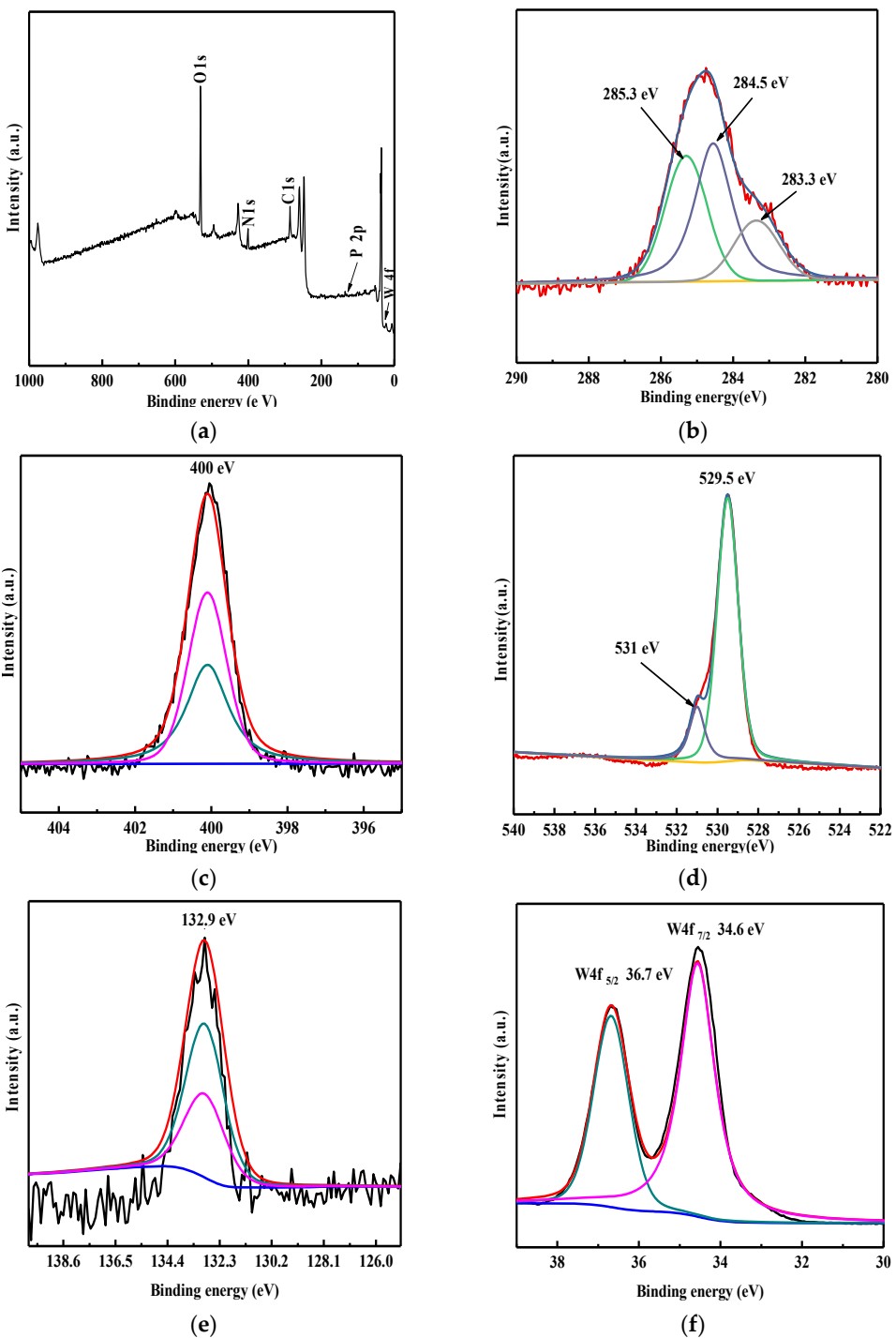

**Figure 2.** XPS spectra of [C$_2$(MIM)$_2$]PW$_{12}$O$_{40}$; (**a**) Survey of the catalyst; (**b**) C 1s; (**c**) N 1s; (**d**) O 1s; (**e**) P 2p; (**f**) W 4f.

Figure 3 shows the results of hydrophilicity and hydrophobicity tests of the catalysts. The instantaneous contact angles of a water droplet on the three POM-DIL catalysts were all less than 90°. The contact angle of [C$_2$(MIM)$_2$]PW$_{12}$O$_{40}$ was also measured with n-octane as the testing droplet. The results showed that the instantaneous contact angle between the oil droplet and the catalyst surface was almost 0. These results indicated that the catalysts have good wettability for both H$_2$O$_2$ and n-octane, which can ef-

fectively improve the utilization rate of the oxidant and the overall desulfurization efficiency.

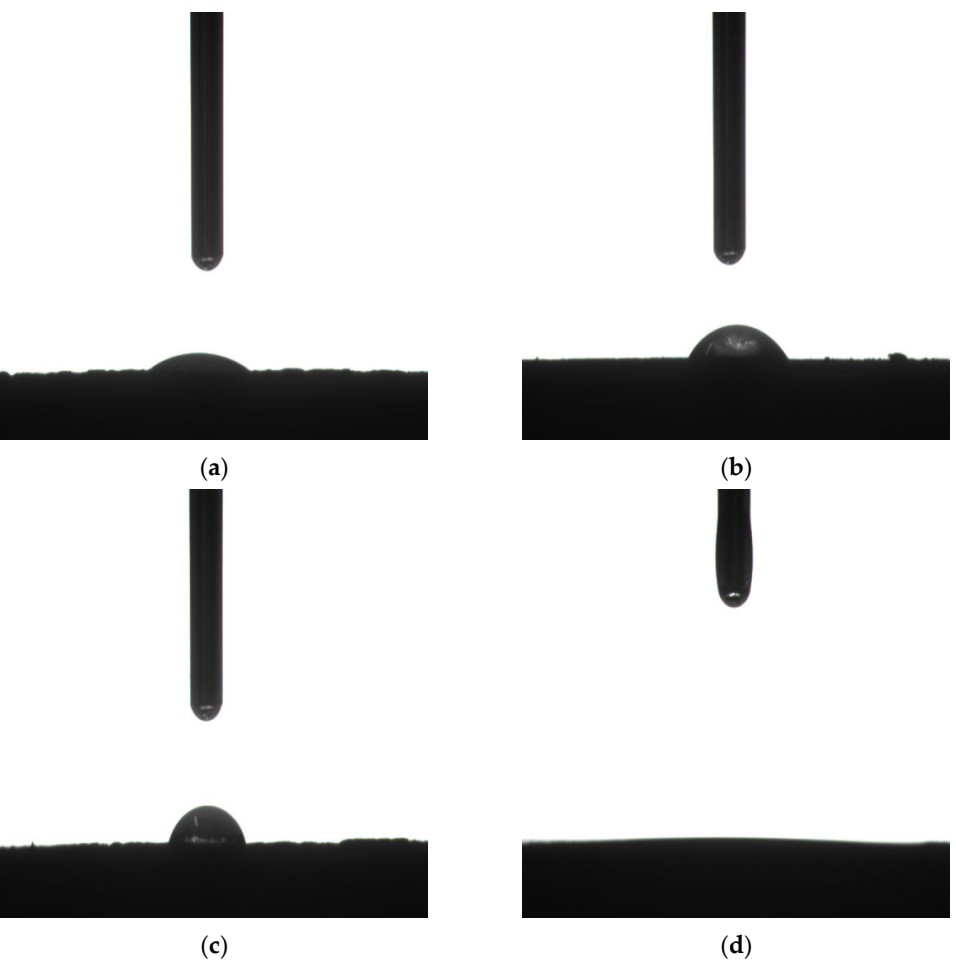

**Figure 3.** Contact angles of water droplets on the surface of the catalyst: (**a**) [C$_2$(MIM)$_2$]PW$_{12}$O$_{40}$; (**b**) [C$_4$(MIM)$_2$]PW$_{12}$O$_{40}$ and (**c**) [C$_6$(MIM)$_2$]PW$_{12}$O$_{40}$. Contact angles of n-octane droplets on the surface of the catalyst: (**d**) [C$_2$(MIM)$_2$]PW$_{12}$O$_{40}$.

### 2.2. Catalytic Activity of Catalyst

Table 3 summarizes and compares the effects of different kinds of POM-IL catalysts on the DBT removal effect in fuel oil under major reaction parameters, such as the H$_2$O$_2$/DBT molar ratio (n(H$_2$O$_2$)/n(S)), reaction temperature and reaction time. The results showed that increasing the length of the carbon chain in catalyst cation has no obvious effect on the desulfurization effect under the same reaction conditions. However, when the volume of cation increased, the desulfurization effect was obviously improved. On one hand, the catalyst had good wettability in the oil phase and H$_2$O$_2$, which can rapidly interaction with H$_2$O$_2$ and oil. When the catalyst was fully contacted with the oxidant, it could decompose more active substances [44], which was more conducive to the removal of sulfide. On the other hand, the large cations had higher aromatic π electron density, which could effectively enhance the π-π interaction between the double imidazole ring and the thiophene ring, thus making POM-DILs have excellent desulfurization performance. Considering the economic factors and desulfurization effect, [C$_2$(MIM)$_2$]PW$_{12}$O$_{40}$ was selected for the subsequent desulfurization research.

The solubility of catalysts in different solvents is an important factor to be considered in their application. Therefore, the solubility of the catalyst in model oil and acetonitrile was tested. According to the results in Table 4, the catalysts were slightly dis-

solved in n-octane and acetonitrile, and tended to be dissolved in acetonitrile with relatively strong polarity. Combined with the desulfurization effect in Table 3, the partial dissolution of the catalyst in solvent had little effect on desulfurization efficiency and oil quality, which can be almost ignored.

**Table 3.** Comparison of different catalysts for removal of dibenzothiophene (DBT) in model fuel.

| Catalyst | Reaction conditions | S-removal (%) | Ref |
|---|---|---|---|
| $[C_2(MIM)_2]PW_{12}O_{40}$ | n(catalyst)/n(S) = 0.025; n(H$_2$O$_2$)/n(S) = 6; 50 °C; 60 min; acetonitrile = 0.5 mL; V(model oil) = 5 mL | 98.4 | This work |
| $[C_4(MIM)_2]PW_{12}O_{40}$ | n(catalyst)/n(S) = 0.025; n(H$_2$O$_2$)/n(S) = 6; 50 °C; 60 min; acetonitrile = 0.5 mL; V(model oil) = 5 mL | 97.0 | This work |
| $[C_6(MIM)_2]PW_{12}O_{40}$ | n(catalyst)/n(S) = 0.025; n(H$_2$O$_2$)/n(S) = 6; 50 °C; 60 min; acetonitrile = 0.5 mL; V(model oil) = 5 mL | 95.5 | This work |
| $[C_4mim]_3PW_{12}O_{40}$ | m(catalyst) = 0.03 g, 60 °C, 30 min, n(H$_2$O$_2$)/n(DBT) = 3 | 11.4 | [45] |
| $[C_8mim]_3PW_{12}O_{40}$ | m(catalyst) = 0.03 g, 60 °C, 30 min, n(H$_2$O$_2$)/n(DBT) = 3 | 10.3 | [45] |
| $[C_{16}mim]_3PW_{12}O_{40}$ | m(catalyst) = 0.03 g, 60 °C, 30 min, n(H$_2$O$_2$)/n(DBT) = 3 | 12.5 | [45] |
| $Cs_{2.5}H_{0.5}PW_{12}O_{40}$ | 60 min; 60 °C; O/S = 15; acetonitrile = 60 mL | 70.5 | [46] |
| $[C_{16}mim]_3PW_{12}O_{40}$ | m(catalyst) = 0.01 g; 60 °C; 1 h; O/S = 3; [Bmim]BF$_4$ = 1 mL | 21.4 | [47] |

**Table 4.** Solubility of catalyst in n-octane and acetonitrile.

| Catalyst | Solvent | Solubility (g/100 g) | Solvent | Solubility (g/100 g) |
|---|---|---|---|---|
| $[C_2(MIM)_2]PW_{12}O_{40}$ | n-octane | 0.0276 | Acetonitrile | 0.0325 |
| $[C_4(MIM)_2]PW_{12}O_{40}$ | n-octane | 0.0193 | Acetonitrile | 0.0275 |
| $[C_6(MIM)_2]PW_{12}O_{40}$ | n-octane | 0.0166 | Acetonitrile | 0.0250 |

Condition: m(catalyst) = 0.005 g; V(solvent) = 5 mL; T=50 °C; t = 60 min.

After the ECODS system was established, the initial reaction conditions were optimized to ensure the best desulfurization efficiency. The effect of reaction temperature on the removal of sulfide DBT from model oil was investigated, and the results are displayed in Figure 4. When the temperature increased from 20 to 80 °C, the desulfurization efficiency showed a trend of increasing first and then decreasing. This was because the oxidation reaction was limited by kinetics and POM-DILs could not effectively catalyze desulfurization at low temperature [48]. With the increase in reaction temperature, the activity of the catalyst and oxidant was gradually enhanced, and the oxidation rate of DBT into DBTO$_2$ was accelerated. However, with the further increase in reaction temperature, the desulfurization efficiency would decrease slightly due to the gradual decomposition of H$_2$O$_2$ and the deactivation of active components [16,49]. Therefore, the best reaction temperature was determined to be 50 °C.

It can be seen from Figure 5, the desulfurization efficiency increased greatly in the initial stage of the reaction. After the reaction for 60 min, the desulfurization efficiency reached the maximum. This may be because the sulfide content in the model oil was highest in the initial reaction stage, so both the extraction rate and oxidation rate were much higher than the other reaction stage. Then, with the increase in reaction time, the overall desulfurization reaction tended to equilibrium, and the desulfurization efficiency did not increase significantly, or even slightly decreased, which was due to the partial volatilization of n-octane in a longer reaction time. Therefore, taking into account the effect of reaction time, the t = 60 min was selected to be used in the rest of the experiments.

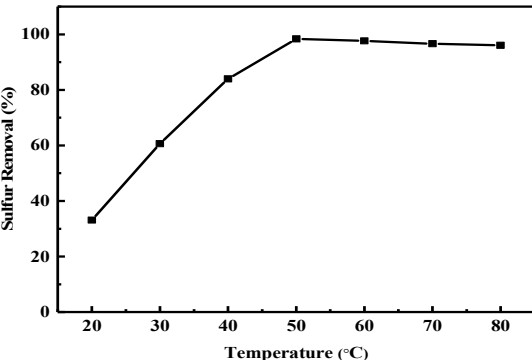

**Figure 4.** Effect of reaction temperature on desulfurization. Reaction conditions: V(oil) = 5 mL; n(catalyst)/n(S) = 0.025; n(H₂O₂)/n(S) = 6; V(acetonitrile) = 0.5 mL; t = 60 min.

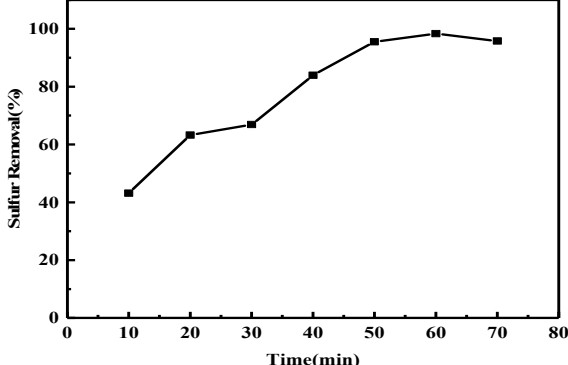

**Figure 5.** Effect of reaction time on desulfurization. Reaction conditions: V(oil) = 5 mL; n(catalyst)/n(S) = 0.025; n(H₂O₂)/n(S) = 6; V(acetonitrile) = 0.5 mL; T = 50 °C.

The influence of the catalyst amount on the desulfurization effect was also considered. As shown in Figure 6, without catalyst, the desulfurization effect was 67.11%. When the molar ratio of catalyst to sulfide was increased to 0.025, the desulfurization effect was increased to 98.35%. According to the reported literature [50], the high catalytic efficiency of $H_3PW_{12}O_{40}$ composites in oxidative desulfurization (ODS) was mainly due to the existence of catalytic active center W=O. Therefore, we speculated that with the increase in the amount of catalyst, the active sites provided for oxidative desulfurization increased, which promoted the effective removal of DBT. When the molar ratio of catalyst to sulfide was further increased, the desulfurization effect did not change obviously. Therefore, 0.025 was a suitable molar ratio of catalyst to sulfide.

The effect of oxidant dosage on the removal of sulfide DBT from model oil was presented in Figure 7. When the desulfurization system was carried out in the absence of $H_2O_2$, the desulfurization efficiency was only about 67.48%. However, when the n($H_2O_2$)/n(S) was increased from 0 to 4, the conversion of DBT was improved significantly. After further increasing the n($H_2O_2$)/n(S) to 6, the removal of DBT could reach 98.35%. According to the stoichiometric reaction, the oxidation of 1 mol DBT to the corresponding sulfones requires 2 mol of $H_2O_2$ [51]. In theory, the excessive amount of $H_2O_2$ was beneficial to fully oxidize DBT to sulfones. In practice, considering the desulfurization efficiency and excessive oxidant may cause oil pollution, n($H_2O_2$)/n(S) = 6 was appropriate.

The effect of the extractant dosage on DBT removal from model oil is shown in Figure 8. When the desulfurization test was conducted without acetonitrile as the extractant, the removal rate of DBT in the model oil reached 69.49%, which was the result of the combination of oxidant and catalyst. With the increase in the dosage of extractant, the desulfurization effect was significantly improved. This was because when the catalyst,

oxidant and extractant were added to the reactor, an environment similar to the emulsion was formed. This environment could effectively increase the contact among the catalyst, oxidant and sulfide, so as to improve the desulfurization efficiency. When the amount of extractant became greater than 0.5 mL, the desulfurization efficiency was slightly improved. Therefore, 0.5 mL extractant was suitable for sulfide extraction.

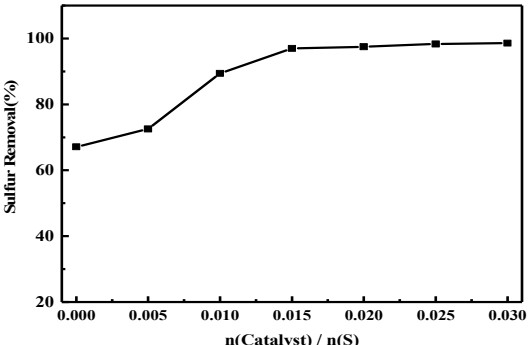

**Figure 6.** Effect of the mole ratio of catalyst to sulfide on desulfurization. Reaction conditions: V(oil) = 5 mL; n(H₂O₂)/n(S) = 6; V(acetonitrile) = 0.5 mL; t = 60 min; T=50 °C.

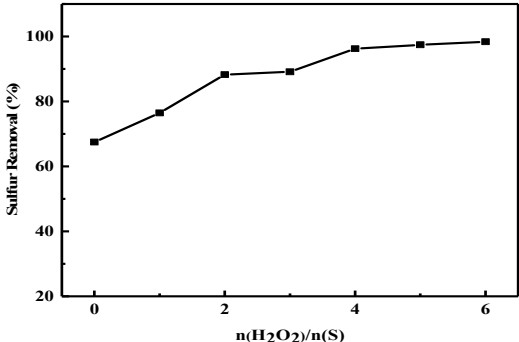

**Figure 7.** Effect of the mole ratio of oxidant to sulfide on desulfurization. Reaction conditions: V(oil) = 5 mL; n(catalyst)/n(S) = 0.025; V(acetonitrile) = 0.5 mL; t = 60 min; T = 50 °C.

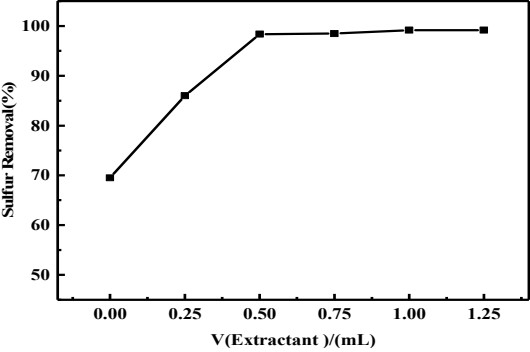

**Figure 8.** Effect of extractant dosage on desulfurization efficiency. Reaction conditions: V(oil) = 5 mL; n(catalyst)/n(S) = 0.025; n(H₂O₂)/n(S) = 6; t = 60 min; T = 50 °C.

Many ILs have been developed which are used as both the catalyst and extractant. Usually, the cation side of ILs influences the extraction ability of this material for DBT removal [38]. Therefore, the extraction ability of DILs was investigated. The results in Table 5 show that the DILs also exhibited a certain extraction ability. Due to the higher aromatic π-electron density of DILs, a stronger π-π interaction could be formed between

the cations of DILs and aromatic sulfides, so DILs have a higher extraction efficiency than monocationic ionic liquids and ordinary organic solvents.

**Table 5.** Effect of different extractants on the removal of DBT.

| Entry | Extractants | Sulfur removal (%) |
|---|---|---|
| 1 | $[C_2(MIM)_2]Cl_2$ | 68.67 |
| 2 | $[C_4(MIM)_2]Cl_2$ | 60.66 |
| 3 | $[C_6(MIM)_2]Cl_2$ | 57.21 |
| 4 | Acetonitrile | 48.37 |
| 5 | Methanol | 31.68 |
| 6 | $BMIMBF_4$ | 55.17 |
| 7 | $BMIMPF_6$ | 56.86 |

Reaction conditions: V(oil) = 5 mL; V(extractant) = 0.5 mL; t = 60 min; T = 50 °C.

### 2.3. Recycling of Catalysts

The recyclability of the POM-DIL catalyst in the reaction system was further investigated from the perspective of economic cost. After the reaction, the upper oil phase was taken for analysis. The solvent in the system was separated by the decanting method, and the catalyst was dried at 100 °C to remove the residual $H_2O_2$, acetonitrile and model oil. Then, the catalyst was reused for the next cycle by the addition of fresh model oil, oxidant and extraction agent. Each group of data was repeated at least three times, and the standard deviation was less than 1.2. As results show in Figure 9, the desulfurization rate was still up to 89.42% after seven cycles of the catalyst. Compared with the catalytic performance of the fresh catalyst, the desulfurization effect was only slightly reduced (<10%), indicating that the catalyst $[C_2(MIM)_2]PW_{12}O_{40}$ had good recycling performance. In addition, the compounds in the oil phase before and after desulfurization were tested by gas chromatography-flame ionization detection (GC-FID). It can be seen from the results in Figure 10 that the oxidation product $DBTO_2$ of DBT was detected in the oil phase after the reaction, which indicated that some oxidation products were still in the oil phase. It could be inferred from the results that with the increase in the number of cycles, the decrease in desulfurization efficiency might be partly due to the presence of oxidation product in the oil phase, which resulted in the blocking of mass transfer in the reaction process. Figure 11 shows the infrared spectrum analysis of the recycled catalyst and fresh catalyst. After recycled tests, the structure of the catalyst was not destroyed, indicating that the catalyst has excellent stability.

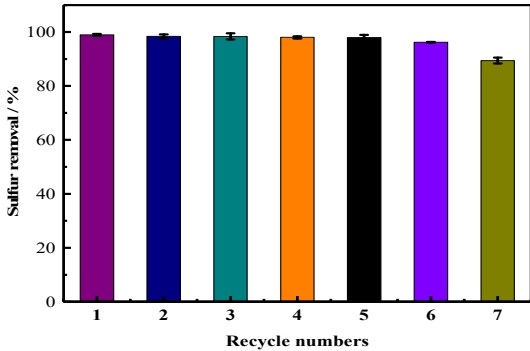

**Figure 9.** Desulfurization efficiency of recycling system. Reaction conditions: V(oil) = 5 mL; n(catalyst)/n(S) = 0.025; $n(H_2O_2)/n(S)$ = 6; t = 60 min; T = 50 °C; V(acetonitrile) = 0.5 mL.

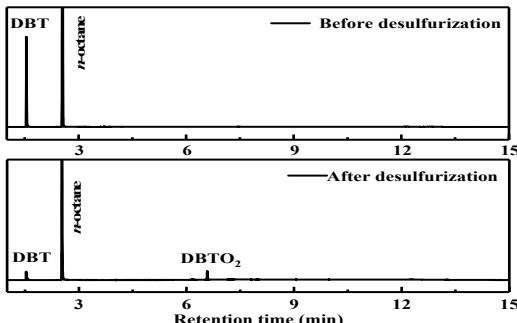

**Figure 10.** GC-FID of the model oil containing DBT before and after desulfurization in the proposed system.

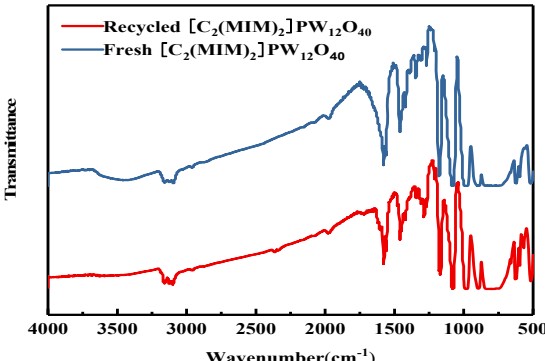

**Figure 11.** FT-IR spectra of fresh and recycled [C$_2$(MIM)$_2$]PW$_{12}$O$_{40}$.

### 2.4. Oxidation Product and Reaction Mechanism

The oxidation product of DBT was further verified by Gas Chromatography-Mass Spectrometry (GC-MS), and the results are shown in Figure 12. From the analysis results, it can be concluded that the oxidation product of DBT in model oil was DBTO$_2$ (m/z = 216.0). Based on our research and the related literature reports [52], we hypothesized the reaction mechanism of ECODS, as shown in Scheme 1. It was assumed that DBT was first extracted into the extraction phase by acetonitrile and POM-DILs with the extraction function. In the process of catalytic oxidation, [PW$_{12}$O$_{40}$]$^{3-}$ in the catalyst was oxidized by H$_2$O$_2$ to the intermediate product [PO$_4$\{W(O)(O$_2$)$_2$\}$_4$]$^{3-}$ with strong oxidation, and H$_2$O$_2$ was activated to form ·OH (the catalytically active O species). Then, by a redox reaction between DBT and the intermediate product and hydroxyl radical, DBT was selectively oxidized to DBTO$_2$, and [PO$_4$\{W(O)(O$_2$)$_2$\}$_4$]$^{3-}$ was reduced to the original [PW$_{12}$O$_{40}$]$^{3-}$. As the reaction proceeded, DBT was continuously oxidized and extracted, and the content of DBT in the oil phase was continuously reduced, while the oxidation product DBTO$_2$ was continuously accumulated in the extraction phase until the end of the reaction.

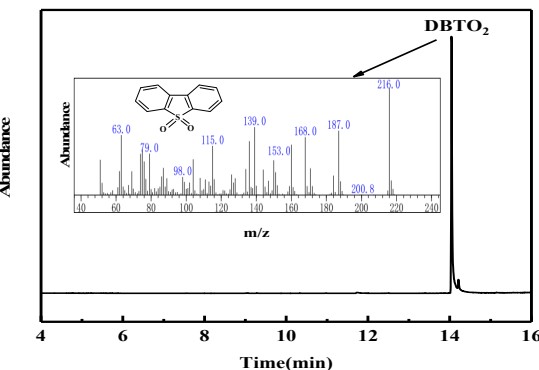

**Figure 12.** Gas Chromatography-Mass Spectrometry (GC-MS) analysis of catalyst after reaction.

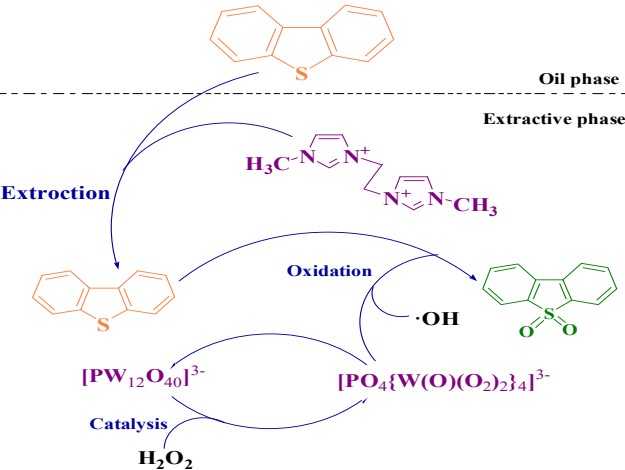

**Scheme 1.** Proposed mechanism for the oxidation of DBT in the ECODS.

## 3. Materials and Methods

### 3.1. Materials

1-Methylimidazole, 1,2-dichloroethane, 1,4-dichlorobutane, 1,6-dichlorohexane, DBT and $H_3PW_{12}O_{40}$ were purchased from MACKLIN (Shanghai, China). Acetone, $H_2O_2$ (30%), acetonitrile, ethanol and n-octane were purchased from Sinopharm Chemical Reagent Co., Ltd. (Shanghai, China). All the reagents and chemicals were directly used in experiments without any purification.

### 3.2. Synthesis of Catalyst

POM-DIL catalysts were prepared by a two-step method. Take $[C_2(MIM)_2]PW_{12}O_{40}$ as an example, and the synthesis steps are shown in Scheme 2. 1-methylimidazole (3.284 g, 0.04 mol) and 1,2-dichloroethane (1.9792 g, 0.02 mol) were charged into a 100 mL round-bottomed flask with a condensation reflux device. Under solvent-free conditions, the reaction mixture was stirred at 90 °C for 2-4 h until white solid appeared. The white solid intermediate product $[C_2(MIM)_2]Cl_2$ was washed repeatedly with acetone to remove nonionic residues and dried at 80 °C in a vacuum for 6 h. Then, $[C_2(MIM)_2]Cl_2$ (0.3946 g, 1.5 mmol) and $H_3PW_{12}O_{40}$ (2.8800 g, 1.0 mmol) were dissolved in 30 mL distilled water and stirred for 2 h at room temperature. After centrifugation and drying, the final catalyst $[C_2(MIM)_2]PW_{12}O_{40}$ was obtained. Preparation of $[C_4(MIM)_2]PW_{12}O_{40}$ and $[C_6(MIM)_2]PW_{12}O_{40}$ was similar to that of $[C_2(MIM)_2]PW_{12}O_{40}$. The results of $^1H$ nuclear magnetic resonance ($^1H$ NMR) spectroscopy (400 MHz, DMSO) characterization were as follows: $[C_2(MIM)_2]PW_{12}O_{40}$: δ 8.98 (s, 2H), 7.72 (s, 2H), 7.57 (s, 2H), 4.65 (s, 4H), 3.86 (s, 6H), 3.37 (s, 6H), 2.52 (s, 2H). $[C_4(MIM)_2]PW_{12}O_{40}$: δ 9.07 (s, 2H), 7.72 (s, 2H), 4.20 (s, 4H),

3.86 (s, 6H), 3.37 (s, 6H), 2.50 (s, 2H). 1.79 (s, 2H). $[C_6(MIM)_2]PW_{12}O_{40}$: δ 9.07 (s, 2H), 7.72 (s, 2H), 4.20 (s, 4H), 3.86 (s, 6H), 3.48 (s, 6H), 2.52 (s, 4H), 1.81 (s, 2H), 1.30 (s, 2H).

**Scheme 2.** Synthetic route of $[C_2(MIM)_2]PW_{12}O_{40}$.

### 3.3. Characterization

$^1$H NMR spectra were obtained on BRUKER AVANCE 400 (Karlsruhe, Germany) using dimethyl sulfoxide (DMSO) as the solvent. Fourier transform infrared spectroscopy (FT-IR) analyses were performed on a Nicolet 6700 FT-IR spectrometer (Thermo Fisher, Massachusetts, USA) using KBr pellets at room temperature. XRD was performed on an Empyrean X-ray diffractometer (PANalytical B.V., Almelo, Netherlands) equipped with Cu-Kα source. The scan speed and step size were 5°/min and 0.02°, respectively. UV-vis spectra were obtained with a UV-vis spectrometer (UVmini-1280, Shimadzu, Suzhou, China) in acetonitrile. TG analyses were carried out on Microcomputer differential thermal balance HCT-3 instrument (Beijing Hengjiu Scientific Instrument Factory, Beijing, China) from 35 to 800 °C, with a heating rate of 10 °C /min in $N_2$ atmosphere. The moisture content was determined on a Karl Fischer Moisture Titrator (870 KF Titrino plus, Heirishau, Switzerland). The main components of KF reagent were $I_2$, $SO_2$, pyridine (buffer) and methanol (solvent). XPS was carried out on ESCALAB250xi (Thermo Scientifle, Massachusetts, USA) with a monochromatic Mg-Kα source with 1487 eV of energy to explore the surface composition. The contact angle tests were conducted on a contact angle instrument (JC2000D, Shanghai Zhongchen Digital Technic Apparatus Co. Ltd., Shanghai, China). The oxidation product of DBT in the model oil was measured by GC-MS (Agilent 5975C, California, USA) and the signals were collected from 4 to 16 min.

### 3.4. Oxidative Desulfurization Process

Model oil containing specific sulfide was selected for the test. In this experiment, DBT (500 mg/L) was used as model oil substrate and n-octane as solvent to form model oil. Firstly, a certain amount of catalyst, $H_2O_2$ (30%) and acetonitrile were added to a 100 mL round bottomed flask containing 5 mL of model oil. Then, the mixture was magnetically stirred for a period of time in a thermostatic water bath. After the reaction, the mixture precipitated for a certain time, the upper oil phase was taken and the sulfide content was determined by GC-FID (VF-1column type; 30 m × 0.25 mm × 0.25 μm; column temperature: 230 °C; the temperature was raised from 100 to 230 °C at the rate of 20 °C /min and kept for 2 min; injection temperature: 300 °C; detector temperature: 320 °C). Each group of data was repeated at least three times. According to the initial and final sulfur content in the model oil, the desulfurization efficiency can be calculated as fol-

lows: S-removal efficiency (%) = $(1-S_1/S_0) \times 100\%$, where $S_0$ is the initial sulfur content in model oil (mg/L); $S_1$ is the final sulfur content in the model oil (mg/L).

## 4. Conclusions

In this study, three kinds of imidazole-based POM-DIL catalysts were synthesized by a two-step method and applied to the removal of DBT from model oil with acetonitrile as the extractant and $H_2O_2$ as the oxidant. Compared with the previous POM-IL desulfurization system, the experimental conditions of the present work were greatly optimized. Under the optimum reaction conditions—n([C$_2$(MIM)$_2$]PW$_{12}$O$_{40}$)/n(S) = 0.025; n($H_2O_2$)/n(S) = 6; V(acetonitrile) = 0.5 mL; T = 50 °C; t = 60 min—the removal rate of DBT reached above 98%. Meanwhile, [C$_2$(MIM)$_2$]PW$_{12}$O$_{40}$ could be reused seven times without obvious catalytic activity loss, and the original Keggin structure of the POM-DIL catalyst was undamaged. The results showed that the catalyst displayed good catalytic activity, stability and recycling performance. This is because the double cation can effectively regulate the interaction between the catalyst and sulfide, oxidant, which has a great impact on enhancing its desulfurization performance. In addition, increasing the carbon chain lengths of the catalysts with short carbon chain displayed no significant effect on the desulfurization efficiency due to the good wettability for both $H_2O_2$ and model oil. Finally, the oxidation product of DBT was proved to be DBTO$_2$ by GC-MS. Detailed analysis of the ECODS mechanism found that the W=O in the catalyst was the active center to activate $H_2O_2$ to generate ·OH, which was beneficial to desulfurization process. This study provided a new reference for the development of an efficient catalyst for catalytic oxidative desulfurization.

**Author Contributions:** Conceptualization, B.H. and J.T.; methodology, B.H.; software, J.L.; validation, B.H.; formal analysis, J.L.; investigation, J.L. and Y.G.; resources, J.L.; data curation, J.L.; writing—original draft preparation, J.L.; writing—review and editing, B.H. and J.T.; visualization, B.H.; supervision, B.H.; project administration, B.H.; funding acquisition, B.H. All authors have read and agreed to the published version of the manuscript.

**Funding:** This research was funded by Hubei Natural Science Foundation Project (2013CKB032) and The Doctoral Foundation Project of Hubei University of Technology (0701).

**Data Availability Statement:** The data presented in this study are available in article.

**Acknowledgments:** We would like to thank the teachers of the school of materials and chemical engineering for their support in the inspection equipment used for experiments.

**Conflicts of Interest:** The authors declare no conflict of interest.

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
