# Peer review of "Polyoxometalate Dicationic Ionic Liquids as Catalyst for Extractive Coupled Catalytic Oxidative Desulfurization"

_catalysts, doi:10.3390/catal11030356_

Round 1

Reviewer 1 Report

The manuscript presents studies on the development of an efficient catalyst for the extractive coupled catalytic oxidative desulfurization of fuel oil by H2O2 and acetonitrile. The novel family of imidazole-based polyoxometalate dicationic ionic liquids (POM-DILs) [Cn(MIM)2]PW12O40 (n=2,4,6) were synthesized by modifying phosphotungstic acid (H3PW12O40) with double imidazole ionic liquid. In my opinion, the manuscript is clear, well-organized and interesting. However, some several parts of manuscript need improved i.e.:

  • Why typical models such as Box-Behnken design or Central composite design were not used to optimization studies?
  • Table 1; Table 2 – what is the difference between processes? (row of columns 1, 2 and 3).
  • Figure 10 - The order of the captions to the chromatograms should be changed.
  • Line 357 – “ppm’” is not official unit - please correct it.
  • What type of GC column was used?
  • More information on the design and parameters of XPS, NMR, FT-IR etc. devices should be included in the paper.
  • Figure 9 - Error bars should be added. How many times were the experiments repeated?

Reviewer 2 Report

The manuscript requires several changes before considering it for publication. These are listed below.

Some paragraphs are unclear and need to be rewritten. For example: p. 3 "However, the smaller specific surface area of POMs (.10 m2/g) results in lower activity [22] and the difficulties in separation and recovery of ILs due to
their liquid properties, which limit their industrial application. "

2. Some of the notations are not explained in the text and might be confusing for the readers. For example:  [Cnmim]PMoV , PhPyBs-PW, etc.

3. The notation of the compounds investigated in this study is confusing, especially when these compounds are compared with others reported in the literature: for example see Table 1, [C2(MIM)2]PW12O40 vs [C4mim]3PW12O40 and so on. Such notation introduces also some confusion in other discussions, for example [Cn(MIM)2]+ (p. 3; the charge should be +2). I suggest to properly formulate the three compounds considering the actual charge of the ions or to use simple 1-3.

4. The language should be consistent "UV-vis spectra" should be used in place of "UV-vis spectrums" (p. 4)

5. It is not properly described how the UV-VIS spectra were recorded (solid-state, solution?; no details are provided in the experimental section).

6. p. 4: the UV-VIS maxima listed are not assigned to the corresponding compounds.  

7. W-Oe-W should be explained, p. 4 line 127.

8. The discussion of the TG analysis is not clearly presented. As it is discussed in the text and presented in Figure 1f, the weight loss between room temperature and 400oC is due to the loss of solvent (water). The loss is quite significant and therefore it is difficult to understand how the authors established the solubility of these products or how they established the correct molar ratio in the different catalytic experiments. These products should be checked for their purity by elemental analysis. The content of water in the samples should be precisely measured. 

9. In the proposed mechanism, the role of the HO. radical is not precisely discussed. 

Round 2

Reviewer 1 Report

Manuscript can be accepted in present form.

Author Response

Thank you very much for your valuable suggestions and comments on our article. It's a great honor to be recognized by you. Thank you again.

Reviewer 2 Report

The authors performed a series of corrections to the original manuscript that now is significantly improved. Yet, the issue with the moisture in the samples is not clearly explained. It is not a matter of physical or lattice water, it is about the quantity in the investigated samples. If this amount is not known precisely then how the authors know the correct molar ratio used in their experiments or the correct solubility of their compounds? As long as the purity of the samples and the content of water are not known, it seems that the results are ambiguous! 

Round 3

Reviewer 2 Report

The authors performed the required changes to their manuscript that can be accepted for publication.